# Navigating the Rare and Dangerous: Successful Clipping of a Superior Cerebellar Artery Aneurysm Against the Odds of Uncontrolled Hypertension

**DOI:** 10.3390/jcm13237430

**Published:** 2024-12-06

**Authors:** Corneliu Toader, Matei Serban, Razvan-Adrian Covache-Busuioc, Mugurel Petrinel Radoi, Ghaith Saleh Radi Aljboor, Luca-Andrei Glavan, Antonio Daniel Corlatescu, Milena-Monica Ilie, Radu M. Gorgan

**Affiliations:** 1Department of Neurosurgery “Carol Davila”, University of Medicine and Pharmacy, 020021 Bucharest, Romania; corneliu.toader@umfcd.ro (C.T.); razvan-adrian.covache-busuioc0720@stud.umfcd.ro (R.-A.C.-B.); petrinel.radoi@umfcd.ro (M.P.R.); ghaith-saleh-radi.aljboor@rez.umfcd.ro (G.S.R.A.); luca-andrei.glavan0720@stud.umfcd.ro (L.-A.G.); antonio.corlatescu0920@stud.umfcd.ro (A.D.C.); milena-monica.ilie0720@stud.umfcd.ro (M.-M.I.); radu.gorgan@umfcd.ro (R.M.G.); 2Department of Vascular Neurosurgery, National Institute of Neurology and Neurovascular Diseases, 077160 Bucharest, Romania; 3Department of Neurosurgery, Clinical Emergency Hospital “Bagdasar-Arseni”, 041915 Bucharest, Romania

**Keywords:** superior cerebellar artery aneurysm, microsurgical clipping, posterior circulation aneurysm, occipito-parietal far-lateral approach, hypertension management, wide-necked aneurysm, cerebrovascular surgery, aneurysm exclusion

## Abstract

**Background/Objectives**: Superior cerebellar artery (SCA) aneurysms are exceedingly rare, posing unique diagnostic and management challenges due to their complex anatomical location within the posterior circulation. The proximity of the SCA to vital structures, such as the brainstem and cerebellum, along with the significant role of poorly controlled hypertension in aneurysm formation, further complicates treatment. This case aims to highlight the surgical approach and management strategies employed in treating a rare SCA aneurysm in a patient with longstanding hypertension. **Methods**: A 68-year-old female presented with an acute onset of severe headache, nausea, and vomiting, later confirmed to be due to a ruptured SCA aneurysm. The patient’s history of poorly controlled hypertension was identified as a major contributing factor to the aneurysm’s development and rupture. Due to the aneurysm’s wide-neck morphology and irregular shape, microsurgical clipping was selected as the treatment method. The occipito-parietal far-lateral approach was employed to gain optimal access to the aneurysm, minimizing the risk to adjacent neurovascular structures. **Results**: Microsurgical clipping successfully excluded the aneurysm while preserving the parent artery. The surgical approach enabled precise aneurysm isolation and ensured no postoperative neurological deficits. The patient recovered well, with no significant complications, and hypertension management was emphasized as a vital element of the patient’s long-term care. **Conclusions**: The surgical technique effectively achieved complete aneurysm exclusion with preservation of the parent artery and no associated neurological deficits. The patient’s recovery was uneventful, highlighting the efficacy of the approach. Long-term management of hypertension remains a critical component to prevent recurrence and ensure sustained outcomes.

## 1. Introduction

Superior cerebellar artery (SCA) aneurysms are exceedingly rare, accounting for less than 1% of all intracranial aneurysms. Detailed case reports are essential for improving understanding of their diagnosis and treatment. Due to the limited number of reported cases in the literature, each case provides valuable insights into effective management strategies. Our case stands out due to its challenging anatomical location and the successful application of microsurgical clipping. This technique involves the use of a surgical microscope to place a specialized clip at the neck of the aneurysm, effectively excluding it from circulation while preserving the integrity of the parent artery. Microsurgical clipping remains a critical option in posterior circulation aneurysms, especially in cases with complex morphologies or wide necks where endovascular techniques may be less effective. Endovascular treatment, in contrast, employs minimally invasive catheter-based approaches to deploy devices such as coils or flow-diverters within the aneurysm sac, promoting thrombosis and preventing rupture. While endovascular techniques are widely used, their limitations in specific cases highlight the continued importance of microsurgical approaches in aneurysm management [1]. In recent years, endovascular treatments have gained prominence in managing posterior circulation aneurysms, including those of the SCA. However, SCA aneurysms, especially those located near the brainstem or cranial nerves, pose considerable risks when treated with coiling or flow-diverting devices due to potential perforator infarction and brainstem ischemia [2].

Despite these advancements, surgical clipping remains a key option, especially for aneurysms with wide necks or complex morphologies that make endovascular occlusion challenging [3].

Traumatic or dissecting aneurysms of the SCA are particularly rare, further complicating diagnosis and management. These aneurysms often present with subarachnoid hemorrhage or, less commonly, ischemic events due to thromboembolic complications. As such, a flexible, multidisciplinary approach is necessary, balancing the risks and benefits of both microsurgical and endovascular methods [4].

## 2. Case Report

A 68-year-old female presented to our clinic with a sudden onset of severe headache, nausea, and vomiting, which began approximately 24 h prior to admission. These symptoms arose abruptly in an otherwise stable clinical state. The patient has a documented history of essential hypertension, which has been poorly controlled despite ongoing treatment. The patient’s antihypertensive treatment regimen included nimodipine at a total daily dose of 480 mg, administered as 120 mg every 6 h for two weeks. Additional antihypertensive agents comprised indapamide SR (1.5 mg) taken once daily in the morning and candesartan (8 mg) administered as half a 16 mg tablet in the evening. Despite this comprehensive therapy, blood pressure control remained suboptimal, significantly contributing to the development and rupture of the superior cerebellar artery aneurysm.

Upon neurological examination, the patient exhibited significant nuchal rigidity, a classic sign of meningeal irritation, and was moderately somnolent with a Glasgow Coma Scale (GCS) score of 10 (Eye response = 3, Verbal response = 3, Motor response = 4), indicating moderate impairment of consciousness. Muscle strength was normal, with an MRC score of 5/5 in both upper and lower limbs. Cranial nerve function was intact, and no abnormalities were detected. Deep tendon reflexes were brisk at +3/4 in all extremities, with no pathological reflexes noted. Sensory examination confirmed normal light touch, pain, and proprioception in all limbs. Due to the patient’s somnolence, a full assessment of coordination was limited, though no signs of dysmetria or ataxia were observed on passive testing. Both Kernig’s and Brudzinski’s signs were positive, reinforcing the clinical suspicion of meningeal irritation.

The acute presentation of headache, along with positive meningeal signs and impaired consciousness, raised concern for a possible central nervous system infection or subarachnoid hemorrhage. Upon admission, the patient’s clinical condition was classified as Hunt and Hess Grade III, reflecting moderate impairment of consciousness, the presence of significant meningeal irritation, and symptoms consistent with a ruptured intracranial aneurysm. This grading highlights the severity of the subarachnoid hemorrhage and further emphasizes the urgent need for surgical intervention.

In this case, our neurosurgical team made the critical decision to bypass cranial computed tomography (CCT) and proceed directly to digital subtraction angiography (DSA), a choice driven by the patient’s life-threatening condition and our team’s extensive expertise in managing similar emergencies. The patient’s symptoms—severe headache, nuchal rigidity, and moderate somnolence—provided unequivocal clinical evidence of subarachnoid hemorrhage, leaving no diagnostic ambiguity. Given the urgency of this life-and-death scenario, where every moment counted, proceeding straight to DSA ensured rapid identification of the aneurysm and expedited surgical planning.

Preoperative cerebral angiography (Figure 1) confirmed the presence of a saccular aneurysm located on the SCA. The aneurysm was clearly visualized in the angiographic series, with lateral, oblique, and anterior–posterior views revealing its close proximity to the basilar artery bifurcation and the posterior cerebral arteries (PCAs). The SCA, a critical vessel supplying the cerebellum and brainstem, places the aneurysm in a high-risk location, with potential involvement of the oculomotor nerve (CN III), given its anatomical course. This could explain the patient’s neurological presentation, including moderate somnolence and meningeal irritation, while also raising the risk of further complications such as brainstem compression or cranial nerve deficits. The aneurysm’s location, in proximity to the pons and midbrain, also poses the threat of cerebellar dysfunction and further neurological deterioration, aligning with the acute onset of severe headache, nausea, and vomiting. The angiographic findings supported the initial clinical concern for a subarachnoid hemorrhage, underscoring the need for urgent neurosurgical intervention.

The 3D angiographic reconstruction (Figure 2) revealed the precise dimensions of the aneurysm located on the superior cerebellar artery. The aneurysm measured 4.69 mm in length and 3.55 mm in width, with a neck diameter of 2.06 mm. These measurements are particularly important when evaluating treatment options, as the size and shape of the aneurysm influence the feasibility of endovascular procedures such as coiling, where smaller neck sizes are generally more amenable. The dome-to-neck ratio, an important factor in deciding between endovascular and surgical intervention, was clearly visible, further aiding in the planning for optimal treatment. The accurate visualization of these dimensions, along with the aneurysm’s anatomical relationship to adjacent vessels, was critical in guiding the neurosurgical team’s decision-making process.

A left-sided occipito-parietal far-lateral approach was utilized to access the SCA aneurysm. This approach, chosen for its optimal exposure to the posterior circulation, began with an occipital craniotomy, allowing for direct yet minimally invasive access to the target region. After incising the dura, the marginal sinus was ligated, and the dura was carefully suspended to create a clear operative field.

Under microsurgical magnification, the cisterna magna was opened, revealing the left superior cerebellar artery along with the vertebral artery at its transition into the intradural space. The proximal-to-distal course of the vertebral artery was followed, and the origin of the SCA was identified. Just distal to this origin, a 4.7 mm saccular aneurysm was located. The aneurysm’s position on the superior aspect of the artery necessitated a careful balance between securing the aneurysm and preserving the surrounding vasculature. A Yasargil clip was applied to the neck of the aneurysm with precision, ensuring the aneurysm was occluded while maintaining the patency of the left SCA. This step required meticulous attention to the anatomy to preserve cerebral blood flow while achieving complete occlusion of the aneurysm. The moderate cerebellar collapse observed during the procedure was managed effectively without compromising the field. Hemostasis was achieved using electrocautery, Surgicel, and tamponade, ensuring a controlled environment for closure. The dura was partially sutured, allowing for postoperative expansion as needed. The occipital bone flap was secured in place using Craniofix, and an epidural drain was positioned to ensure adequate drainage and prevent complications such as hematoma formation. Finally, the incision was closed in anatomical layers, ensuring proper alignment for optimal healing.

Operating on a ruptured aneurysm presented a significant challenge, as the risk of rebleeding during the procedure was ever-present. The fragile state of the aneurysm and the surrounding vascular structures demanded a controlled and deliberate dissection. The presence of subarachnoid blood added complexity, requiring constant suctioning to maintain a clear operative field. To mitigate the risk of rebleeding, meticulous surgical techniques were employed, ensuring minimal manipulation of the aneurysm until the Yasargil clip was positioned. The clip was applied with precision, immediately securing the ruptured aneurysm and halting further bleeding.

Post-clipping, the surgical field was carefully inspected under high-magnification microscopy to confirm complete occlusion of the aneurysm and the patency of the superior cerebellar artery. This visual confirmation, combined with the absence of active bleeding and stable hemodynamic conditions throughout the procedure, provided assurance of a successful outcome. Hemostasis was achieved efficiently, with no intraoperative complications, underscoring the effectiveness of the surgical strategy in managing this high-risk case.

The postoperative CT scan (Figure 3) confirms a successful outcome following the aneurysm clipping procedure. The images demonstrate that the Yasargil clip is securely positioned at the neck of the aneurysm on the left superior cerebellar artery, with no evidence of compression or compromise to the parent vessel. The surrounding brain tissue shows no signs of new hemorrhage or ischemic changes, and the previously noted cerebellar collapse has remained stable without progression. Ventricular size is within normal limits, with no signs of hydrocephalus or cerebrospinal fluid obstruction. Additionally, the bone flap is well-fixed, with no complications related to the craniotomy site. The absence of postoperative complications, such as rebleeding or infarction, alongside the stable positioning of the clip, indicates a smooth recovery process. Clinically, the patient has shown significant improvement, with a marked recovery of consciousness and resolution of preoperative symptoms, such as somnolence and meningeal irritation. The absence of complications, such as rebleeding or ischemia, combined with the secure positioning of the clip and the stable postoperative neurological examination, supports the conclusion of a highly successful intervention.

The patient’s neurological improvement, coupled with the imaging findings, strongly suggests an uncomplicated recovery and favorable long-term prognosis. At the time of discharge, the patient exhibited an excellent neurological recovery. Her GCS score had returned to 15, indicating full restoration of consciousness and cognitive function. Neurological examination revealed no focal deficits, and she was fully independent in her activities of daily living, with a modified Rankin Scale (mRS) score of 0, reflecting complete functional recovery without residual symptoms.

Postoperative blood pressure control remained a critical focus of care. The patient continued her preoperative antihypertensive regimen. She was counseled on the importance of strict adherence to her treatment plan to prevent future cerebrovascular events and was provided with personalized advice on lifestyle modifications, including a balanced diet and regular physical activity.

## 3. Discussion

SCA aneurysms are exceedingly rare, representing less than 1% of all intracranial aneurysms. Their rarity, combined with the complex anatomical location of the SCA within the posterior circulation, poses significant challenges for diagnosis and management. The SCA courses through an area critical to vital functions, home to multiple essential neural structures, including the brainstem and cerebellum. Its proximity to other significant arteries, such as the posterior cerebral artery and the vertebral-basilar system, further complicates the treatment of aneurysms in this region. The unique vascular anatomy of this region requires a highly individualized and meticulous approach to treatment to balance the risks and benefits effectively [2].

The role of hypertension in aneurysm formation and rupture is well-documented, particularly in the context of posterior circulation aneurysms. Hypertension exerts significant hemodynamic stress on the arterial walls, particularly in high-flow regions such as the SCA, where vessels are exposed to sustained high pressures. Over time, this increased pressure leads to the degradation of the tunica media and elastic lamina, which weakens the vessel wall and makes it more prone to aneurysmal dilation and rupture. Effective management of hypertension has been shown to play a critical role in reducing the incidence and progression of aneurysms, especially in patients with underlying risk factors [5]. In our patient, longstanding poorly controlled hypertension likely contributed significantly to the development and eventual rupture of the aneurysm. This underscores the importance of strict blood pressure control, particularly in patients who are predisposed to aneurysm formation due to systemic conditions like hypertension. Managing hypertension as a modifiable risk factor is essential not only in the prevention of aneurysm formation but also in improving postoperative outcomes and reducing the risk of recurrence [2]. In this case, the patient’s poor blood pressure control contributed to the aneurysm’s progression, highlighting the need for comprehensive management of hypertension alongside surgical treatment.

The decision to pursue microsurgical clipping was influenced by several factors, most notably the aneurysm’s wide-neck morphology and irregular shape. Endovascular techniques, such as coiling and stent-assisted procedures, while widely used, are often less effective in treating wide-necked aneurysms due to the increased risk of incomplete occlusion, coil migration, or residual aneurysmal filling. In contrast, microsurgical clipping offers a more definitive solution, allowing for direct visualization of the aneurysm and precise manipulation of the neck, which ensures complete exclusion of the aneurysm while preserving the parent vessel. This approach is particularly advantageous in posterior circulation aneurysms, where the risk of recurrence is higher when treated with endovascular techniques alone [1]. In the present case, the occipito-parietal far-lateral approach was chosen due to its ability to provide optimal exposure to the aneurysm while minimizing the risk to adjacent critical neurovascular structures, such as the brainstem and cranial nerves. This approach is particularly beneficial for deep-seated aneurysms in the posterior circulation, offering enhanced visualization and enabling precise aneurysm isolation [6]. Studies have shown that this surgical approach is associated with favorable neurological outcomes, with lower rates of postoperative complications compared to other approaches [1]. By providing better access to the aneurysm, the far-lateral approach allowed for the successful exclusion of the aneurysm without postoperative neurological deficits in our patient.

The management of hypertension remains crucial in preventing both the formation and rupture of aneurysms, particularly in high-stress vascular territories like the posterior circulation. In our patient, poorly controlled hypertension played a pivotal role in the development and clinical presentation of the aneurysm. Numerous studies have shown that long-term blood pressure management is essential for reducing the risk of rupture and improving overall outcomes in patients with intracranial aneurysms. In this case, the patient’s hypertension likely played a pivotal role in both the development and clinical presentation of the aneurysm. This underscores the need for comprehensive management of systemic conditions, such as hypertension, to mitigate the risk of rupture [7].

While this case report demonstrates the successful management of a ruptured superior cerebellar artery aneurysm using a microsurgical clipping approach, we acknowledge the limitations inherent to single-patient studies. The findings and conclusions presented here are based on a specific clinical scenario and cannot be generalized to all patients with similar aneurysms. Factors such as anatomical variations, comorbid conditions, and institutional expertise may influence the outcomes of such procedures. Furthermore, the absence of comparative data limits the ability to assess the approach against alternative treatment modalities, such as endovascular techniques. Future studies involving larger patient cohorts or multicenter data are necessary to validate the efficacy and reproducibility of this approach in diverse clinical settings. Despite these limitations, this case contributes valuable insights into the management of rare posterior circulation aneurysms and underscores the importance of individualized treatment planning.

## 4. Conclusions

SCA aneurysms are particularly notable due to the rarity of such aneurysms and their complex anatomical location within the posterior circulation. The patient’s poorly controlled hypertension was a key factor in both the aneurysm’s formation and rupture, emphasizing the critical role of managing systemic risk factors in such cases.

Microsurgical clipping was selected as the optimal treatment due to the aneurysm’s wide-neck morphology, and the occipito-parietal far-lateral approach provided the necessary exposure to safely exclude the aneurysm while minimizing risks to surrounding neurovascular structures. The successful outcome, with no postoperative neurological deficits, further reinforces the effectiveness of this approach for complex posterior circulation aneurysms.

This case highlights the need for a tailored, multidisciplinary approach to treating rare aneurysms like those of the SCA. It contributes to the limited literature on these challenging aneurysms and emphasizes the importance of precise surgical techniques combined with rigorous control of hypertension to achieve favorable clinical outcomes.

## Figures and Tables

**Figure 1 jcm-13-07430-f001:**
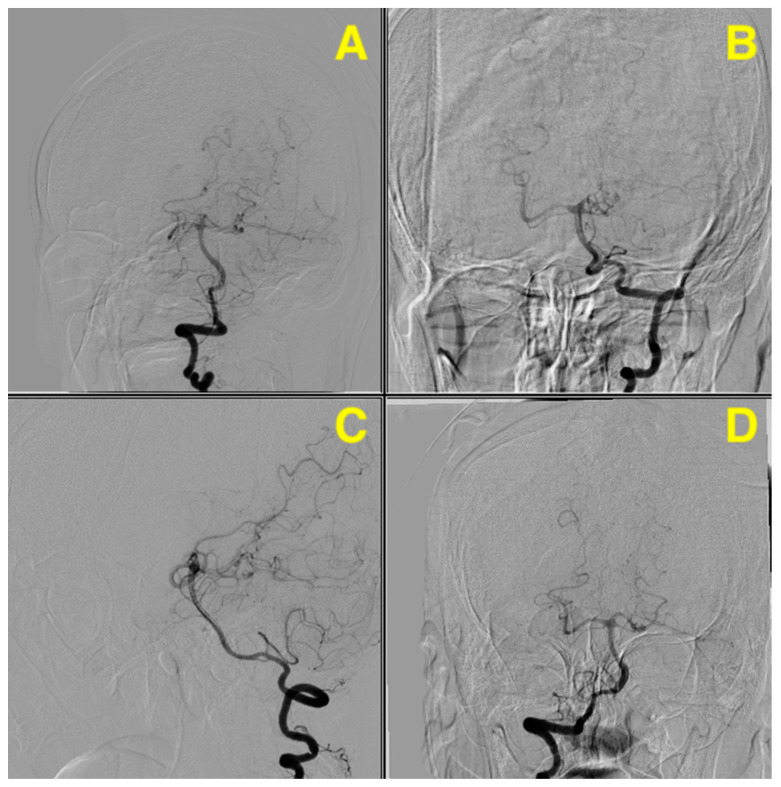
The preoperative angiography demonstrates a ruptured aneurysm located on the middle portion of the left SCA, oriented laterally and posteriorly, with a neck measuring approximately 2 mm in diameter. The maximal dimensions of the aneurysmal sac are approximately 4.7 mm in length and 3.4 mm in width. The vascular imaging outlines the aneurysm’s proximity to the basilar artery bifurcation and its relationship to the surrounding posterior circulation vessels. The lateral projection (**A**) highlights the aneurysm’s orientation and size relative to the parent vessel, emphasizing its lateral and posterior direction, with no apparent filling defects or thrombus within the aneurysmal sac. The anteroposterior view (**B**) delineates the aneurysm’s position along the superior cerebellar artery and its anatomical relationships with the basilar artery and posterior cerebral arteries, clearly defining the vascular bifurcation anatomy and aiding in surgical planning. The oblique projection (**C**) provides additional depth perception of the aneurysm’s spatial orientation, confirming its posterior and lateral projection from the SCA, and offers a detailed understanding of the aneurysm’s dome-to-neck ratio, which is crucial for selecting the surgical approach. The rotational view (**D**) emphasizes the three-dimensional architecture of the aneurysm, providing comprehensive insights into its morphology and the involvement of adjacent vascular structures. These angiographic findings underscore the complexity of the aneurysm’s location, necessitating meticulous planning for surgical clipping. The precise measurements and detailed visualization provided by the angiography were instrumental in guiding the decision to proceed with a left-sided occipito-parietal far-lateral approach, ensuring optimal exposure and minimizing risks to critical neurovascular structures.

**Figure 2 jcm-13-07430-f002:**
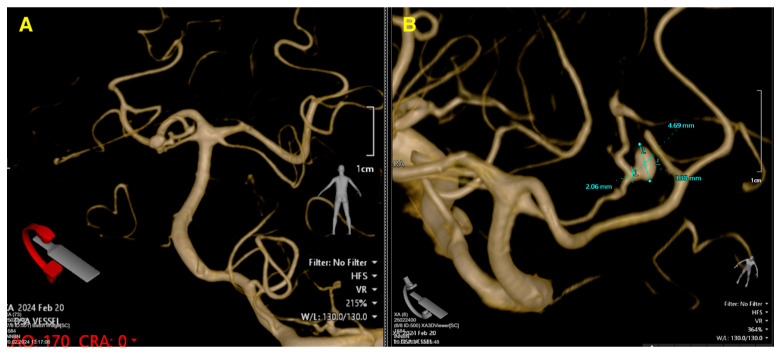
Pre-op 3D angiographic reconstruction. (**A**) offers an overview of the posterior circulation anatomy, emphasizing the aneurysm’s location and its spatial relationship to the basilar artery bifurcation. (**B**) provides precise measurements of the aneurysm, with a dome size of 4.69 mm, a width of 3.55 mm, and a neck diameter of 2.06 mm. The high-resolution 3D imaging highlights the aneurysm’s morphology, including its lateral and posterior orientation, and the vascular structures surrounding it.

**Figure 3 jcm-13-07430-f003:**
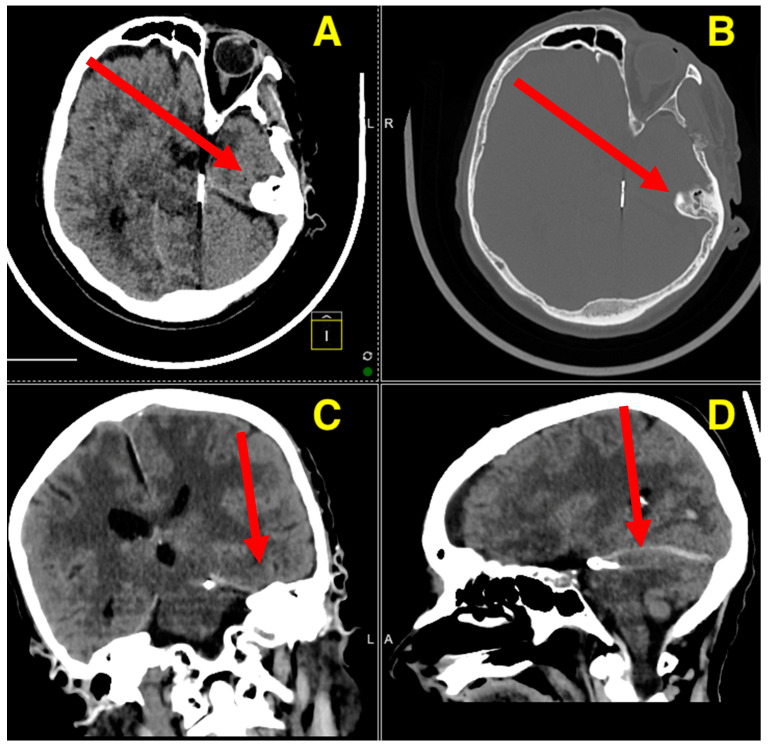
Post-op control CT scan. The axial view (**A**) clearly demonstrates the secure placement of the Yasargil clip at the neck of the aneurysm, with no evidence of residual aneurysmal filling or new hemorrhagic events. The surrounding brain parenchyma shows no signs of ischemic damage, indicating effective hemostasis and preservation of cerebral perfusion. The bone window in the axial plane (**B**) confirms the intact positioning of the occipital bone flap, which was secured with Craniofix, with no evidence of complications, such as fractures, dislodgement, or hematoma formation, at the craniotomy site. The coronal view (**C**) highlights the absence of midline shift, ventricular enlargement, or hydrocephalus, reflecting stable intracranial pressure and normal cerebrospinal fluid dynamics. The cerebellar and brainstem structures appear anatomically intact and unaffected by the surgical intervention. The sagittal reconstruction (**D**) further confirms the precise placement of the clip and the complete exclusion of the aneurysm, with no evidence of residual filling or recurrence. Additionally, there are no signs of mass effect or compression on adjacent structures.

## Data Availability

The data presented in this study are available on request from the corresponding author.

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
