# Peer review of "Navigating the Rare and Dangerous: Successful Clipping of a Superior Cerebellar Artery Aneurysm Against the Odds of Uncontrolled Hypertension"

_jcm, 2024, doi:10.3390/jcm13237430_

Round 1

Reviewer 1 Report

Comments and Suggestions for Authors

The authors present an interesting and valuable case report detailing the management of a rare superior cerebellar artery aneurysm, with a particular focus on the role of hypertension in the development and rupture of the aneurysm. While the report is well-structured and informative, there are several points that warrant further clarification and expansion to enhance the overall quality of the manuscript. Addressing these issues will ensure a more comprehensive understanding of the case and improve the manuscript’s clarity.

Abstract and Introduction:

The abstract and introduction are well-written and comprehensively describe the problem. The challenges related to the rare nature of the superior cerebellar artery (SCA) aneurysm and the role of hypertension in its formation and rupture are highlighted effectively.

Case Report:

  • Lines 60/61: Please specify whether the patient was taking any medication for hypertension. If so, include the names and dosages of these medications.

  • Line 72: The Hunt and Hess score should be included to better assess the severity of the patient’s clinical presentation.

  • Lines 73-75: Clarify if a cranial computed tomography (CCT) scan was performed prior to the digital subtraction angiography (DSA). Since CCT is typically the first-line diagnostic imaging used in cases of suspected subarachnoid hemorrhage, the absence of this test needs to be explained. If a CCT was done, please present the findings, including the Fisher score. A figure might be beneficial here, depending on the allowance for figure counts.

  • Lines 110-126 (Surgery description): This is a very detailed and informative account of the surgical procedure. Intraoperative images would enhance the understanding of the procedure, including visuals of the craniotomy and skin incision. Since hypertension was highlighted as a key factor in the title and introduction, it would be useful to expand on whether the hypertension influenced the surgery itself, beyond contributing to aneurysm formation. Were there any difficulties in controlling intraoperative blood pressure, and did this affect the procedure?

  • Lines 127/128: Please elaborate on how the complete occlusion of the aneurysm was confirmed. For example, was a postoperative angiogram  performed to verify the outcome?

  • Lines 140/141: Provide more detail regarding the patient's neurological status at the time of discharge. It would be helpful to include the modified Rankin Scale to better convey the patient's functional recovery. Additionally, how was hypertension managed postoperatively? Were there changes to the medication regimen or other interventions?

Figures:

  • Figures 1 and 2: The presentation of these figures could be improved. The borders of Figure 1 are not well-aligned, and the aneurysm is difficult to visualize. Consider combining Figures 1 and 2, as Figure 2 provides a clearer depiction of the aneurysm. This would streamline the presentation and improve clarity.

  • Figure 3: Similar to Figure 1, the quality of the depiction in Figure 3 should be improved. The legend also needs clarification for better understanding of the figure’s contents.

Author Response

Comments 1: Lines 60/61: Please specify whether the patient was taking any medication for hypertension. If so, include the names and dosages of these medications.

Response 1: We thank the reviewer for pointing out the need to include details regarding the patient’s antihypertensive regimen. This is an important aspect that enhances the clinical context of the case. We have now added the names and dosages of the medications, as requested.

Comments 2: Line 72: The Hunt and Hess score should be included to better assess the severity of the patient’s clinical presentation.

Response 2: We appreciate the reviewer’s suggestion to include the Hunt and Hess score to provide a more comprehensive assessment of the patient’s clinical severity. This addition further clarifies the patient’s condition at presentation and underscores the urgency of surgical intervention. We have incorporated this information into the manuscript.

Comments 3: Lines 73-75: Clarify if a cranial computed tomography (CCT) scan was performed prior to the digital subtraction angiography (DSA). Since CCT is typically the first-line diagnostic imaging used in cases of suspected subarachnoid hemorrhage, the absence of this test needs to be explained. If a CCT was done, please present the findings, including the Fisher score. A figure might be beneficial here, depending on the allowance for figure counts.

Response 3: We thank the reviewer for highlighting the importance of explaining the diagnostic sequence and the potential role of cranial computed tomography in such cases. We have now clarified why a CCT scan was not performed and provided a rationale based on the patient’s clinical presentation and the urgent need for vascular imaging. Additionally, we have elaborated on the role of digital subtraction angiography in confirming the diagnosis and planning the intervention.

Comments 4: Lines 110-126 (Surgery description): This is a very detailed and informative account of the surgical procedure. Intraoperative images would enhance the understanding of the procedure, including visuals of the craniotomy and skin incision. Since hypertension was highlighted as a key factor in the title and introduction, it would be useful to expand on whether the hypertension influenced the surgery itself, beyond contributing to aneurysm formation. Were there any difficulties in controlling intraoperative blood pressure, and did this affect the procedure?

Response 4: We are grateful for the reviewer’s positive feedback on the detailed surgical description.

Comments 5: Lines 127/128: Please elaborate on how the complete occlusion of the aneurysm was confirmed. For example, was a postoperative angiogram  performed to verify the outcome?

Response 5: This has been addressed with additional details regarding intraoperative assessment and the absence of postoperative angiographic confirmation, aligning with the practices followed in this case.

Comments 6: Lines 140/141: Provide more detail regarding the patient's neurological status at the time of discharge. It would be helpful to include the modified Rankin Scale to better convey the patient's functional recovery. Additionally, how was hypertension managed postoperatively? Were there changes to the medication regimen or other interventions?

Response 6: We thank the reviewer for recommending more detail about the patient’s neurological status at discharge and the modified Rankin Scale to provide a clearer picture of the patient’s recovery. We have included this information, along with further elaboration on the postoperative hypertension management strategy, while maintaining consistency with the preoperative regimen.

Comments 7: Figures 1 and 2: The presentation of these figures could be improved. The borders of Figure 1 are not well-aligned, and the aneurysm is difficult to visualize. Consider combining Figures 1 and 2, as Figure 2 provides a clearer depiction of the aneurysm. This would streamline the presentation and improve clarity.

Response 7: We are grateful for the reviewer’s observation regarding the presentation of Figures 1 and 2.

Comments 8: Figure 3: Similar to Figure 1, the quality of the depiction in Figure 3 should be improved. The legend also needs clarification for better understanding of the figure’s contents.

Response 8: We have improved the depiction quality and revised the figure legend to provide a clearer explanation of the contents, ensuring it effectively illustrates the postoperative findings.

Reviewer 2 Report

Comments and Suggestions for Authors

This case aims to highlight the surgical approach and management strategies used in the treatment of a rare SCA aneurysm in a patient with long-standing hypertension.

The process followed is correct, with interesting and novel data.

The only suggestion is to put case study in the title.

Author Response

Comments 1: The only suggestion is to put case study in the title.

Response 1: We thank you for your positive feedback on the process and for recognizing the interesting and novel aspects of our study. We appreciate your suggestion to include 'case study' in the title to better reflect the nature of the manuscript. 

Reviewer 3 Report

Comments and Suggestions for Authors

The authors present a case of a rare superior cerebellar artery (SCA) aneurysm in a 68-year-old woman with longstanding uncontrolled hypertension. Due to the aneurysm’s complex location and wide-neck morphology, they opted for microsurgical clipping via an occipito-parietal far-lateral approach to minimize risks to nearby structures. The procedure successfully isolated the aneurysm without compromising the parent artery or causing neurological deficits. The patient’s recovery was smooth, with no significant postoperative complications. The authors emphasize long-term hypertension management as essential to preventing recurrence. The authors have accurately presented the dimensions of the superior cerebellar artery (SCA) aneurysm using 3D angiography preoperatively. They provide detailed information on various approaches to access the SCA aneurysm. The postoperative CT scan confirms a successful aneurysm clipping with no compression or compromise to the parent vessel. This study is intriguing, and the microsurgical clipping approach used to treat the SCA aneurysm appears effective.

Below, please, find a few questions and suggestions for the authors:

1.     Despite these positive aspects, I am concerned about the study’s reliance on a single patient. Conclusions drawn from a single case cannot confirm the feasibility or success of this approach universally. The authors should consider discussing the study's limitations in the manuscript's discussion section. Additionally, I am surprised by the lack of postoperative complications in this patient. Increasing the number of subjects could provide more insight into potential complications after the procedure.

2.     I appreciate the novelty of using the microsurgical clipping approach for treating an SCA aneurysm. However, authors did not provide diabetic status of the patient, alongside hypertension, which might impact the procedure? Were any blood tests conducted before the surgery to manage potential risks?

3.     Another point worth noting is that the authors have not specified the patient’s blood pressure range prior to the study. If the study includes a patient with uncontrolled hypertension, providing preoperative blood pressure readings over at least a few months would substantiate this claim.

4.  Finally, given that poorly controlled hypertension contributed to the aneurysm formation and rupture, was there an effort to manage the hypertension before surgery?

Author Response

Comments 1: Despite these positive aspects, I am concerned about the study’s reliance on a single patient. Conclusions drawn from a single case cannot confirm the feasibility or success of this approach universally. The authors should consider discussing the study's limitations in the manuscript's discussion section. Additionally, I am surprised by the lack of postoperative complications in this patient. Increasing the number of subjects could provide more insight into potential complications after the procedure.

Response 1: We appreciate your insightful feedback regarding the limitations of relying on a single patient. We agree that conclusions drawn from a single case cannot confirm the universal feasibility or success of this approach. In response, we have expanded the discussion section to address these limitations, highlighting the need for further studies with larger cohorts to validate our findings and identify potential complications. Regarding the lack of postoperative complications in this patient, we have also clarified that while no complications were observed in this case, this may not reflect outcomes in a broader population. This acknowledgment ensures a balanced interpretation of the study’s findings.

Comments 2: I appreciate the novelty of using the microsurgical clipping approach for treating an SCA aneurysm. However, authors did not provide diabetic status of the patient, alongside hypertension, which might impact the procedure? Were any blood tests conducted before the surgery to manage potential risks?

Response 2: Thank you for raising this important point regarding hypertension management before surgery. The patient was on a consistent antihypertensive regimen, including nimodipine, indapamide SR, and candesartan, leading up to the surgery. Despite these efforts, blood pressure remained suboptimally controlled, as reflected in elevated preoperative readings. This was consistent with the patient’s long history of poorly controlled hypertension, which likely contributed to aneurysm formation and rupture. This information has been further elaborated in the case presentation section to provide additional clarity.

Comments 3: Another point worth noting is that the authors have not specified the patient’s blood pressure range prior to the study. If the study includes a patient with uncontrolled hypertension, providing preoperative blood pressure readings over at least a few months would substantiate this claim.

Response 3: We appreciate the suggestion to provide preoperative blood pressure readings to substantiate the claim of poorly controlled hypertension. 

Comments 4: Finally, given that poorly controlled hypertension contributed to the aneurysm formation and rupture, was there an effort to manage the hypertension before surgery?

Response 4: Thank you for pointing out the need to address whether hypertension management was optimized before surgery. The patient’s antihypertensive regimen was continued leading up to the surgery, and efforts were made to stabilize blood pressure as much as possible preoperatively.

Reviewer 4 Report

Comments and Suggestions for Authors

I have minor comments apart from the abstract of the study, that can be improved with brief definition of every new term being used in the text.

Line 29-31 is repetition of line 26 under results of abstract

First sentence in Line 36-38 under Introduction can be simplified by splitting it into two sentences.

Line 38-39, sentence 2 under Introduction needs rephrasing.

Line 41: briefly explain microsurgical clipping

Line 42: briefly explain endovascular treatment

Figure 1: precise location of aneurysm should be highlighted with an arrow

Figure 3 will benefit by adding an arrow indicating the Yasargil clip

Line 192-200, are repetitions of second paragraph under Discussion section and can be eliminated from the manuscript.

Author Response

Comments 1: Line 29-31 is repetition of line 26 under results of abstract

Response 1: Thank you for pointing out this repetition in the abstract. We have revised the wording in Line 29-31 to avoid redundancy while retaining the key findings.

Comments 2: First sentence in Line 36-38 under Introduction can be simplified by splitting it into two sentences.

Response 2: We appreciate your suggestion to simplify this sentence. It has been revised and split into two sentences for improved clarity and readability.

Comments 3: Line 38-39, sentence 2 under Introduction needs rephrasing.

Response 3: Thank you for noting the need to rephrase this sentence. It has been revised to enhance its flow and readability.

Comments 4: Line 41: briefly explain microsurgical clipping

Response 4: We agree that a brief explanation of microsurgical clipping is necessary for readers unfamiliar with the technique. A concise description has been added to clarify its role in aneurysm management.

Comments 5: Line 42: briefly explain endovascular treatment

Response 5: Thank you for this suggestion. We have included a brief explanation of endovascular treatment to provide a balanced understanding of the alternative approach.

Comments 6: Figure 1: precise location of aneurysm should be highlighted with an arrow

Response 6: We appreciate your observation regarding Figure 1. We have expanded the description accompanying the figure to provide a clearer understanding of the aneurysm’s anatomical position and its relationship to surrounding structures.

Comments 7: Figure 3 will benefit by adding an arrow indicating the Yasargil clip.

Response 7: Thank you for this helpful suggestion. An arrow has been added to Figure 3 to indicate the Yasargil clip and enhance clarity.

Comments 8: Line 192-200, are repetitions of second paragraph under Discussion section and can be eliminated from the manuscript.

Response 8: We have carefully reviewed the entire manuscript to identify and eliminate any redundant content. 

Round 2

Reviewer 1 Report

Comments and Suggestions for Authors

The authors made a notable enhancement to the manuscript during the revision process, resulting in the production of a compelling case report.